# The Quality of Menu Offerings in Independently Owned Restaurants in Baltimore, Maryland: Results from Mixed-Methods Formative Research for the FRESH Trial

**DOI:** 10.3390/nu16101524

**Published:** 2024-05-18

**Authors:** Shuxian Hua, Anna Claire Tucker, Sydney R. Santos, Audrey E. Thomas, Yeeli Mui, Veronica Velez-Burgess, Lisa Poirier, Lawrence J. Cheskin, Mika Matsuzaki, Stacey Williamson, Uriyoan Colon-Ramos, Joel Gittelsohn

**Affiliations:** 1Department of International Health, Johns Hopkins Bloomberg School of Public Health, Baltimore, MD 21205, USA; atucke31@jh.edu (A.C.T.); ssanto13@jhu.edu (S.R.S.); ymui1@jhu.edu (Y.M.); vvelezb1@jh.edu (V.V.-B.); lpoirie4@jhmi.edu (L.P.); mmatsuz2@jhu.edu (M.M.); swill363@jh.edu (S.W.); jgittel1@jhu.edu (J.G.); 2Department of Health, Behavior, and Society, Johns Hopkins Bloomberg School of Public Health, Baltimore, MD 21205, USA; athom203@jhmi.edu; 3Department of Nutrition and Food Studies, George Mason University, Fairfax, VA 22030, USA; cheskin@jhu.edu; 4Department of Medicine, Johns Hopkins University School of Medicine, Baltimore, MD 21205, USA; 5Department of Global Health, Milken Institute School of Public Health, George Washington University, Washington, DC 20052, USA; uriyoan@gwu.edu

**Keywords:** food away from home (FAFH), mixed-methods, healthy food, independently owned restaurants, chronic disease, intervention, dietary guidelines, stakeholder engagement, food deserts, health equity

## Abstract

(1) Background: Independently owned restaurants (IORs) are prevalent in under-resourced racial and ethnic minority communities in the US and present a unique setting for public health nutrition interventions. (2) Methods: We conducted 14 in-depth interviews with IOR owners in Baltimore about their perceptions of healthy food, and customers’ acceptance of healthier menus and cooking methods and concurrent observations of the availability of healthy options on their menus. Qualitative data were coded and analyzed using ATLAS.ti. Observations were analyzed with statistical analysis performed in R. (3) Results: Owners perceived non-fried options, lean proteins, and plant-based meals as healthy. While open to using healthier cooking fats, they had mixed feelings about reducing salt, adopting non-frying methods for cooking, and adding vegetables and whole grains to the menu, and were reluctant to reduce sugar in recipes and beverages. Only 17.5% of 1019 foods and 27.6% of 174 beverages in these IORs were healthy, with no significant differences in the healthfulness of restaurant offerings within low-healthy-food-access/low-income neighborhoods and those outside. (4) Conclusion: Healthy options are generally scarce in Baltimore’s IORs. Insights from owners inform future interventions to tailor healthy menu offerings that are well-received by customers and feasible for implementation.

## 1. Introduction

Over the past three decades, the United States has experienced a steady increase in food consumption from food-away-from-home (FAFH) or prepared-food sources [1]. The role of FAFH in contributing to obesity, cardiovascular diseases, and all-cause mortality has been well documented [2,3]. A recent systematic review revealed that individuals who frequently consume FAFH have higher daily intakes of energy, as well as total and saturated fats and sodium [4]. Because the 2020–2025 Dietary Guidelines for Americans (DGA) suggest that “the nutrient density and healthfulness of what people eat or drink is determined by the preparation methods of food”, regardless of their sources [5], interest has been growing in implementing interventions through FAFH settings, including both national chains and independently owned restaurants (IORs), to promote healthy diets [6,7].

African American communities encounter a higher burden of some chronic diseases compared to other racial and ethnic populations [8]. This health disparity is attributed to upstream social determinants, such as structural racism, which profoundly impact food access and lead to higher food insecurity among racial and ethnic minority communities [9]. Research indicates that African American communities, regardless of income, as well as low-income neighborhoods in general, have limited access to healthy foods that meet the DGA compared to primarily white, higher-income communities [10]. Baltimore, with a majority-African-American population (61.2%) [11], exhibits higher rates of adult obesity (36.4%), diabetes (12.6%), and high blood pressure (35.6%) compared to national averages [12]. In Baltimore, the primary prepared-food sources in low-income areas are IORs, which typically lack a variety of healthy food items, such as vegetables and whole grains [13]. The high concentration of these restaurants in Baltimore’s Healthy Food Priority Areas (HFPAs), defined by the Baltimore City government as “areas with a low average Healthy Food Availability Index score (0–9.5), median household income at or below 185% of the Federal Poverty Level, over 30% of households without a vehicle available, and supermarkets located more than a quarter of a mile away” [14], has driven local researchers to implement community-based interventions to promote healthy eating over the past decade [15,16].

Nationwide, current restaurant interventions focus primarily on increasing the availability of healthy options, promoting such options, and implementing menu labeling [17,18]. However, to date, these interventions have not been shown to significantly impact customers’ consumption patterns and health outcomes [19,20]. Meanwhile, the lack of involvement from restaurant stakeholders in the early development stages of these interventions may well compromise their effectiveness and feasibility. Previous studies have shown that engaging stakeholders in research can improve the acceptability, feasibility, and relevance of the study to its intended audiences [21]. However, only two studies have been published that seek restaurant stakeholders’ perspectives on the acceptability of healthy option promotion strategies [22,23]. This highlights a significant gap in understanding the perspectives of stakeholders about improving the healthfulness of menu offerings in their restaurants to enhance diet quality.

To address this knowledge gap, we conducted mixed-methods formative research in Baltimore, Maryland, for a randomized controlled trial (RCT) entitled Focus on Restaurant Engagement to Strengthen Health (FRESH). The RCT aims to promote healthy eating within low-income, racial and ethnic minority urban communities in the Baltimore City and Washington DC metropolitan areas. The utility of formative research lies in providing evidence of what works and what does not, helping investigators understand the studied context and develop interventions tailored to the target audience [24]. This study seeks to answer three research questions:

(1) What is the current availability of healthy foods and beverages in IORs in Baltimore? And how does this availability vary according to neighborhood healthy food access and income level?

(2) What do owners of Baltimore IORs consider healthy foods on their menus?

(3) What are owners’ perceptions of customer reactions to potential healthy changes in their menus and cooking methods?

## 2. Materials and Methods

### 2.1. Study Design

This mixed-methods study was conducted as formative research for the FRESH trial and spanned from June to December 2023. The research phase comprised three components: (1) in-depth interviews (IDIs) with owners of IORs in Baltimore, (2) an evaluation of restaurant food environments using the Nutrition Environment Measures Survey in Restaurants (NEMS-R, April 2007) [25], including a menu analysis following the DGA for 2020–2025 and the healthy meal guidelines for restaurant nutrition performance standards [5,26], and (3) back-of-house observations. For the present analysis, we reviewed the transcripts of IDIs and performed a healthfulness assessment of menus collected for restaurant food environment evaluation.

### 2.2. Recruitment Strategy

We sought to recruit a diverse sample of IORs. We started by recruiting owners of IORs who had participated in IDIs for a related project. To ensure the range of sample was broad enough to fulfill our research objectives, we later adopted purposive sampling. We utilized a list from Google Maps that included multiple types of IORs (e.g., sit-down, carryout, and food market vendors) across different neighborhoods, including HFPAs and low-income neighborhoods (defined as having more than 15% of the population living below the poverty level). ArcGIS Online software (June 2022, version 1.9) was used to determine if an IOR was located within an HFPA [27]. Neighborhood demographic data (e.g., % population living below the poverty level, % African American residents) were obtained from the US Census Bureau [11]. Two graduate research assistants (SH and AET) began the recruitment process with in-person visits to restaurants, distributing recruitment flyers and verbally introducing the research project (Appendix A). If owners expressed interest, we scheduled a time to return to conduct data collection. Recruitment continued until data saturation for the IDIs was achieved. Data saturation was determined by data collectors collectively when no new themes or information emerged from subsequent interviews [28].

### 2.3. Eligibility Criteria

The study participants were required to be over 18 years of age and to own an IOR located within Baltimore’s city limits. National chains or franchises, as well as specialty stores (e.g., bakeries, coffee shops, smoothie bars), were excluded. Eligible IORs were included regardless of their neighborhood profiles (i.e., income level, whether they were in HFPAs or not).

### 2.4. In-Depth Interviews

IDIs (*n* = 14) were conducted with eligible IOR owners by two trained research assistants (SH and AET) from June to November 2023. The interviews were held in person, via Zoom, or over the telephone based on participants’ preferences, and were conducted in English. Informed consent was obtained from all participants. Questions covered the general background of the restaurant, menu development, customers’ interest in healthier foods, and general perceptions of the project (Appendix A). The interviews were recorded on mobile phones, transcribed verbatim, and cleaned of filler words (e.g., like, um, you know, etc.) by AET and SH. Participants were compensated with a USD 40 gift card.

### 2.5. Menu Healthfulness Assessment

Paper menus from the 14 IORs were collected following the completion of IDIs. If paper menus were unavailable, we utilized menu information posted on delivery apps or IOR websites. To evaluate the availability of healthy food and beverage options across the 14 IOR menus, we conducted a menu healthfulness assessment through observation. We utilized the food group framework established by the DGA for 2020–2025 and the healthy meal guidelines from a 2012 conference in Santa Monica, California, as our standard for identifying healthy restaurant offerings [5,26]. In accordance with these guidelines, food items qualified as healthy if they were non-fried vegetables and/or fruits; whole-grain options; non-fried fish/seafood; non-fried, skinless, white-meat poultry; non-fried beans; and non-fried tofu [26]. Healthy beverages include calorie-free beverages (e.g., water, tea, coffee, flavored water, sparkling water), low-calorie sweetened beverages (e.g., diet sodas), or those that are significant nutrient contributors, such as 100% juice without added sugar [5].

The process for counting food and beverage items followed the NEMS-R protocol [25]. All menu items, including various preparation options as explicitly stated on the menu (e.g., fried or baked, chicken or beef, in barbecue or honey mustard sauce, regular or diet) were identified as separate items. Items offered in standard portion sizes (e.g., 10, 15, or 20 chicken wings) or as part of a combo meal with another food (e.g., a sandwich with fries, counted as two separate items) were tallied only once; combo meals were not double-counted. Customized items (e.g., build your own pizza) were also counted once. We classified individual foods from multiple sections of the menu, including appetizers and side dishes. We excluded desserts (e.g., sliced cakes, packaged sweet snacks) because these items are not categories listed in the NEMS-R protocol. Tap water was included in the beverage category, while alcoholic beverages were excluded.

### 2.6. Researcher Characteristics and Reflexivity

The majority of our research team consists of students and faculty from the School of Public Health who identify as white or Asian and are not originally from Baltimore. Our team members have lived and worked in Baltimore for durations ranging from 2 to 30 years. Recognizing our positionality in Baltimore, a city with a predominantly African American population, we intentionally increased the number of African American-owned restaurants in our sample to better understand the perspectives of owners who more closely identify with the community they serve. This approach was taken to more effectively capture the community’s perspectives and to assess the acceptability of the intervention through the lens of local food practices.

### 2.7. Ethical Considerations

All participant information, including the names, locations of restaurants, and personal details of owners, were anonymized prior to analysis using assigned codes. Photographic, audio, and textual data were digitized and stored in a password-protected document management system, Microsoft OneDrive. The study was approved by the Johns Hopkins School of Public Health Institutional Review Board (IRB00024491) on 5 April 2023.

### 2.8. Data Analysis

IDI Analysis: We applied thematic analysis and adopted an inductive coding approach [29]. Three research team members developed a codebook after independently reviewing the interview transcripts and engaging in iterative group discussions. This initial codebook was structured around three key components of the planned intervention: suppliers, customers, and restaurants. The coding process was completed using ATLAS.ti software (December 2023, version 23.4). Each analyst independently coded half of the transcripts; then, the team collectively reviewed the codebook for relevance and consistency with the data. Any discrepancies in codes and definitions were addressed at this stage. For this analysis, SH further refined the codebook based on the IDI guide to ensure alignment with the research questions. The final stage of analysis involved synthesizing the coded data into overarching themes, which focused on IOR owners’ perceptions of healthy foods and customers’ reactions to potential modifications in menus and cooking methods.

Menu Analysis: We calculated the percentage of items classified as “healthy” relative to the total number of food or beverage items for each food or beverage group. We also calculated the percentage of restaurants offering these healthy options. Using R software (June 2022, version 4.2.1), we conducted a Student *t*-test to assess whether differences in the percentage of healthy menu items were statistically significant across neighborhoods with varying levels of healthy food access and income. The significance threshold was set at a *p*-value of 0.05. Restaurants were categorized according to their location in neighborhoods with low healthy food access or low income, referred to as HFPA/low-income, and those in areas without these characteristics, termed non-HFPA/low-income.

## 3. Results

### 3.1. IOR Characteristics

Fourteen IORs were included in the study (Table 1). Fifty-seven percent were restaurants providing no seating options for customers. Forty-three percent were situated in HFPA/low-income neighborhoods. Forty-three percent served neighborhoods with predominantly (≥80%) African American populations, and fifty-seven percent of the restaurants were African American-owned.

### 3.2. Availability of Healthy Foods and Beverages in Baltimore’s IORs

A total of 1019 food items and 174 beverage items were identified from the fourteen Baltimore IOR menus, including 473 foods and 52 beverages compiled from menus of six IORs located in HFPA/low-income neighborhoods. Few menu offerings were classified as “healthy”, with only 17.5% of food items and 27.6% of beverage items meeting the standards in total. IORs in HFPA/low-income neighborhoods had a higher proportion of healthy beverage items (28.8% vs. 27.0%) but a lower proportion of healthy food items (13.5% vs. 20.9%) compared to IORs in non-HFPA/low-income neighborhoods. Interestingly, a higher percentage of IORs within HFPA/low-income neighborhoods offered at least one healthy menu option compared to those in non-HFPA/low-income areas. Specifically, 100% of IORs in HFPA/low-income areas offered healthy food and 83.3% offered healthy beverages, whereas only 75% of IORs in non-HFPA/low-income areas offered healthy food and beverages. In terms of specific food groups, healthy seafood (5.1%), poultry (5.0%), and dark-green vegetables (4.1%) were the most commonly available in general, with ten to twelve IORs offering these options. In contrast, soy products (0.3%) were rarely available, with only two IORs providing those items, both of which were in non-HFPA/low-income neighborhoods. The most frequently available healthy beverages were sugar-free beverages (17.8%), provided by ten IORs.

Table 2 presents a comparison of the healthy menu offerings between two samples of IORs in HFPA/low-income and non-HFPA/low-income neighborhoods. There were no statistically significant differences in the percentage of healthy menu offerings between IORs within HFPA/low-income neighborhoods and those outside these areas, suggesting a consistently low proportion of healthy menu offerings across various neighborhood food environments and socioeconomic statuses.

### 3.3. Owners’ Perceptions of Healthy Foods

IOR owners defined healthy foods in three ways: (1) non-fried options, (2) lean protein, and (3) plant-based meals.

#### 3.3.1. Non-Fried Options

Owners unanimously agreed that using non-frying cooking methods, such as grilling, baking, boiling, and sautéing, contributes to the healthfulness of a meal. Examples frequently mentioned by owners included grilled meats and sautéed vegetables.

“Well, I would definitely say the sautéed broccoli, the salmon, the grilled fish.” —Sit-down-restaurant owner in a non-HFPA/low-income neighborhood

Meats were more commonly prepared in fried forms compared to vegetables. While many restaurants offered both non-fried and fried options, frying was typically the default cooking method for meat unless the menu specified “grilled” or customers requested preparation in other ways. Despite this trend, one owner operating a restaurant in a low-income neighborhood used grilling as the standard cooking method for chicken dishes, aiming to offer customers a healthier diet.

“I have filet of chicken. I have chicken breast. They’re not breaded unless they ask for certain, we do have a certain chicken breast that we bread and fry, but for most of our chicken breasts we put on the grill.”—Carryout restaurant owner in a low-income neighborhood

#### 3.3.2. Lean Protein

Lean protein, especially lean meats such as poultry, seafood, and fish, emerged as a common theme for healthy food options. Besides serving as a standalone item like a grilled chicken entrée, it was also considered healthy when incorporated as an ingredient, even if processed or combined with seasonings such as mayonnaise. For instance, both turkey ham sandwiches and chicken salad were considered healthy choices by a sandwich restaurant owner and were described as popular among customers.

“Now the other thing is, we have our cold subs turkey, which is very healthy. Tuna, chicken salad, we make a lot of that, and we sell that every other day we make it.”—Sit-down-restaurant owner in a non-HFPA/low-income neighborhood

Seafood and fish have been pivotal elements of Baltimore’s culture, with the blue crab serving as a city symbol. Many local IORs are specialized crab houses that offer a variety of preparation methods, including steaming, boiling, and frying, or serving them as ready-to-eat meals like crab cakes. Other popular selections included tuna, salmon, whiting, lake trout, mussel, oysters, and shrimp. Owners emphasized their crucial roles in providing healthy options for customers with dietary restrictions around poultry and red meat.

“See the best thing that’s healthy… I’d stick with the salmon, really, we actually have salmon. That’s a good format. Yellow rice and salmon, the crab and the shrimp if you want a pescetarian diet.”—Carryout restaurant owner in a non-HFPA/low-income neighborhood

#### 3.3.3. Plant-Based Meals

For some owners, plant-based meals were deemed healthy. Such meals primarily consist of rice, grains, and vegetables. One owner provided a rationale behind this perception, pointing out that some meats offered in restaurants are processed and, therefore, not as healthy.

“More healthy on the menu? For me, looking at it from my side of view, I would say the oatmeal, the fruit, the veggie omelet, the Greek omelet, it’s good, it doesn’t have any meat, because a lot of people, some of the meat I know, it’s processed, so it can’t be healthy to any of us, even though we like it.”—Sit-down-restaurant owner in a non-HFPA/low-income neighborhood

Among all plant-based foods, vegetables were the most frequently mentioned. They were presented in a multitude of ways: in salads, as side dishes, or as elements of compound dishes such as veggie omelets. The nutritional value of vegetables, particularly their high fiber content, was emphasized as a key reason for their perceived health benefits in a diet.

“I think definitely a fair share of vegetables, important nutrients, I guess. Maybe salad. Super high fiber.”—Sit-down-restaurant owner in an HFPA/low-income neighborhood

When it comes to rice and grains, there were perceptions that these foods may increase the risk of diet-related chronic diseases, such as diabetes, due to their high carbohydrate content. One African American owner serving a low-income neighborhood specifically raised concerns over the consumption of rice and bread, both of which are refined grain products, and their impact on the health outcomes of racial and ethnic minority populations.

“Because our neighborhood is African American and Hispanic. And two of the populations with the highest rate of diabetes. And that’s because of the diet. Hispanics have a lot of rice and carbs as well in their diet. And we like things fried, and we like bread and things like that.”—Carryout restaurant owner in a low-income neighborhood

In contrast, products made from whole-grain flour (including wheat, rye, oats, barley, and more) are often considered healthier than products made from refined wheat flour, such as white bread.

“We have [whole] wheat bread, rye bread, which is very healthy compared to [white bread], you can put a turkey, a tuna, a chicken salad, that stuff.”—Sit-down-restaurant owner in a non-HFPA/low-income neighborhood

### 3.4. Owners’ Perceptions of Customers’ Reactions to Healthier Changes

Table 3 presents a summary of IOR owners’ perceptions, attitudes, reasons given, and suggestions regarding customers’ reactions to potential menu and cooking method modifications aimed at improving the healthfulness of diet. Themes are organized by each component of change as outlined in the questions from the IDI guide.

#### 3.4.1. Use Healthier Cooking Fats: Positive Reaction

IOR owners were generally open to the idea of switching from cooking oils high in saturated fats and/or trans fats, typically found in butter and margarine, respectively, to those containing monounsaturated fats, such as olive oil and canola oil. Half of the owners believed that customers would be receptive to the change in oils, given that the difference in taste and flavor is generally minimal.

“I do not even think they would be able to tell.”—Carryout restaurant owner in an HFPA/low-income neighborhood

However, they noted cost as a significant consideration since oils of a higher nutritional quality tend to be more expensive.

“I think that would be a pretty good idea, actually, as long as we were able to keep using it, I guess, and fit it into our budget.”—Sit-down-restaurant owner in an HFPA/low-income neighborhood

#### 3.4.2. Reduce Salt: Mixed Reaction

Owners expressed mixed opinions about customer receptiveness to reduced salt. Most of the owners believed that reducing salt in recipes was feasible. Six out of fourteen owners reported that some customers already request lower salt levels in their orders. One owner associated this preference with health consciousness, particularly concerning high blood pressure.

“Usually, the only thing that people are usually concerned about here, when they order, they’ll say ‘no salt.’ Usually, no salt. Maybe no mayonnaise on certain things, but it’s usually no salt. I guess because of high blood pressure.”—Carryout restaurant owner in a non-HFPA/low-income neighborhood

In response to these customer preferences, several restaurants, including those in HFPA/low-income neighborhoods, have adapted by reducing added salt in their offerings. They compensated for reduced saltiness by enhancing flavors with alternative seasonings, such as herbs and spices.

“So, I do get some customers that don’t want any salt. They still want the flavor, so we’ll just do herbs. And I do have, I actually do have a separate container inside where I just have onion powder, garlic powder. Let’s say, I have some dried thyme. So, like I said, just the spices and the herbs, no salt.”—Carryout restaurant owner in an HFPA/low-income neighborhood

However, resistance to reduced salt existed, with a subset of owners observing customer preference for traditional salt seasoning over alternatives. One owner who served an HFPA/low-income neighborhood pointed out that customers would likely notice a change in taste.

“But I think for seasoning like salt, they probably would notice, and they would probably prefer to have salt over another seasoning.”—Sit-down-restaurant owner in an HFPA/low-income neighborhood

#### 3.4.3. Adopt Non-Frying Cooking Methods: Mixed Reaction

As previously mentioned, most restaurants, regardless of neighborhood profile, offered both non-fried and fried options, with the ability to accommodate custom requests even when frying was the standard method of preparation. Owners generally did not consider increasing the availability of non-fried options to be a substantial challenge.

“I think we could probably do a lot of more grilling instead of frying. I don’t think that would really be a big issue.”—Carryout restaurant owner in an HFPA/low-income neighborhood

Nonetheless, opinions were divided regarding customers’ predicted tendency to choose non-fried over fried food. Some owners noted a distinct trend among their health-conscious clientele, who preferred food prepared without frying.

“There are people who want to know ‘do you fry it or do you broil it?’ And when we say we broil it, they’re happy. So, there are people out there who, they do want to eat healthy.”—Carryout restaurant owner in a non-HFPA/low-income neighborhood

Conversely, some owners reported a prevalent customer behavior pattern, with a significant portion of their clientele consistently choosing fried dishes over non-fried alternatives. One African American owner operating in an HFPA/low-income neighborhood noted that their customers have a strong preference for fried food.

“We offer, like I said, we offer the barbecue and the baked. So they can put it together how they want to. But their choice is always fried. Fried and fries.”—Carryout restaurant owner in an HFPA/low-income neighborhood

#### 3.4.4. Offer More Vegetable Items: Mixed Reaction

Restaurant owners serving HFPA/low-income neighborhoods conveyed mixed perspectives on how customers might react to more non-fried vegetable items as opposed to less nutritious options such as fries and processed meat. Some owners observed a growing interest in vegetarian dishes and believed that providing more vegetable options could meet customer demands.

“As of late, I’ve had people asking for vegetarian type dishes. We have people that ask for salad. We used to do a salad. We don’t do a fresh garden salad. We don’t do that anymore. But I have had people ask for that. And I think they would prefer that over the French fries.”—Carryout restaurant owner in an HFPA/low-income neighborhood

Conversely, a view more commonly expressed among restaurant owners highlighted that only a small fraction of their clientele routinely included vegetables in their meat-based entrées. Some customers may forgo them entirely.

“30%, they’ll come in and get a salad, or a side of collard greens with some yams or vegetable to go with their meat. 70% of them just come in and get some meat and go.”—Carryout restaurant owner in an HFPA/low-income neighborhood

For strategies to incorporate more vegetable options on the menu, one African American owner offered a suggestion that extends beyond altering food items; he proposed introducing juices as a preliminary step to familiarize customers with vegetables.

“The first thing you start off to transfer people in that way is… I say through juice, through juicing. If you want to get people on greens and kale and all that, just hide it through the beverage.”—Carryout restaurant owner in a non-HFPA/low-income neighborhood

#### 3.4.5. Increase Whole-Grain Options: Mixed Reaction

The discussion regarding whole grains primarily centered on bread options for sandwiches. Among the fourteen owners interviewed, five already offered alternatives such as rye or wheat bread alongside the more commonly used white bread; three of these owners operated within HFPAs. Nonetheless, they observed greater customer preference for refined wheat products.

“It’s not a lot. We have a few. I probably count them on one hand, how many customers we have like that. Because most of them like either a bun or a white bread.”—Carryout restaurant owner in an HFPA/low-income neighborhood

The variety of whole-grain products is currently limited, as no interviewed restaurants provided options beyond whole-wheat bread or oatmeal. However, several owners were interested in diversifying their whole-grain offerings. One African American owner, serving an HFPA/low-income neighborhood, saw potential in quinoa as a favored whole-grain choice for customers.

“I’m thinking about maybe adding a plant-based option. Probably something involving quinoa because I love quinoa. It’s a good option because it’s not super heavy like rice, but it’s very flavorful, it’s a grain. So, I like quinoa, I think people would probably take it.”—Carryout restaurant owner in an HFPA/low-income neighborhood

#### 3.4.6. Reduce Sugar: Negative Reaction

The consensus among owners indicated a reluctance to reduce sugar or switch to sugar alternatives due to a strong customer preference for sweet flavors, particularly in beverages, and this was reported by owners serving both inside and outside HFPAs.

“In here, this surrounding area, I mean, I keep it the same. You’ll definitely know if somebody didn’t make it right, they’ll complain that there’s not enough sugar. You don’t know how many people complain about putting more syrup in the shaved ice. And that is literally all sugar.”—Food market vendor in a non-HFPA/low-income neighborhood

The challenge, as reported by a restaurant in an HFPA, is that customers may be sensitive to the taste difference when sugar is reduced or substitutes are used.

“I feel like it’s kind of hard to change [sugar] in the drinks because they would probably notice.”—Sit-down-restaurant owner in an HFPA/low-income neighborhood

Another reason is the cultural significance of sugary drinks, like fruit punch and half-and-half (a blend of sweet tea and lemonade), in Baltimore. Most owners recognize a “comforting” food culture embraced by locals, epitomized by customers seeking “cheat day meals” that typically include a fried chicken box and a half-and-half to achieve psychological and physical satisfaction. They would strongly oppose any reduction in sugar in these beverages, despite acknowledging the detrimental impact on health these options have.

“We’d probably get some negative feedback about that. I can tell you, our half-and-half is… a diabetes freeway. Yeah, it’s not going to happen. Because they will just stop getting it.”—Carryout restaurant owner in an HFPA/low-income neighborhood

Despite these challenges, some owners have had success with cane sugar alternatives. For instance, an African American-owned restaurant found that using brown sugar required less quantity to achieve the desired sweetness.

“My butternut squash pie, I use brown sugar, okay? And the brown sugar is because it hits better than the white sugar, alright? So, if I use brown sugar, I can use a cup and a half. I don’t have to use four cups of white sugar.”—Sit-down-restaurant owner in a non-HFPA/low-income neighborhood

Moreover, an owner from a diner proposed reducing sugar intake by controlling refills of sweetened teas.

“And we do give free refills with the sweet tea. Maybe what we can do is not give him refills on the sweet tea and give free refills on the unsweetened tea.”—Sit-down-restaurant owner in a non-HFPA/low-income neighborhood

## 4. Discussion

In this study of IORs in Baltimore, Maryland, we sought to understand restaurant owners’ perceptions regarding healthy foods and receptivity to changes in menus and cooking methods that could improve the healthfulness of offerings. While nearly all restaurants in our sample offered healthy foods, less than one-quarter of the foods and beverages offered in restaurants met established standards for healthy meals. This trend is not unique to Baltimore IORs; a recent domestic nutrition profile study of fast food chain restaurants revealed that less than 20% of menu items were classified as healthy as of 2016 [30]. No significant differences were observed in our study concerning the availability of healthy offerings, irrespective of whether restaurants were in HFPA/low-income neighborhoods. A similar finding emerged from a restaurant healthfulness study conducted in a county in Northeast Ohio, demonstrating that a less healthy restaurant environment prevails not only in food deserts but also in middle- and high-income neighborhoods [31]. These findings indicate that policies and interventions aimed at increasing the availability of healthy offerings in restaurant settings are warranted.

Owners’ perceptions of healthy food on their menus, including “non-fried options”, “lean protein”, and “plant-based meals”, align with the health criteria set by the DGA 2020–2025 and restaurant nutrition performance standards for healthy meals. This alignment indicates a degree of nutrition knowledge when owners make decisions on menu offerings. However, there was a noticeable knowledge gap regarding the healthfulness of processed meats. While owners regarded high-fat processed meats, such as bacon and sausage, as unhealthy, they considered lean meats (e.g., turkey bacon, chicken salad, deli ham) as generally healthy regardless of processing. This viewpoint is misaligned with recommendations from the World Health Organization, which has classified both high-fat and lean processed meats as carcinogenic [32]. Our findings align with a recent study in New York City, where Latin American restaurant owners identified healthy menu options as green salads, vegetarian alternatives, and seafood [23]. In contrast, several studies in Asian countries focused on cooking methods, such as sodium reduction in IORs, found that owners paid attention to the healthfulness of less strongly flavored foods rather than the overall composition of a meal [33,34].

Regarding owners’ attitudes toward intervention strategies, switching to healthier cooking fats emerged as the most acceptable method to improve diet quality, although the greater cost of oils that are high in monounsaturated fats (e.g., olive oil) posed a potential challenge. Instead, options like canola and vegetable oils were deemed more feasible for future intervention due to their cost comparability with the fats currently used in restaurants [35]. Reducing salt received mixed responses from restaurant owners. However, the strategy of blending salt with a variety of seasonings, herbs, and spices to maintain the dishes’ flavor may be feasible, as some restaurants already do so. In fact, a study that modified menu items from six chain restaurants and conducted blind taste tests found that the majority of menu items with slight reductions in sodium (e.g., reductions in mayonnaise, creamy dressings, cheese, fried onions, bacon, salt, and seasoning) were still acceptable to participants [36]. Increasing offerings of non-fried, plant-centered options also received mixed responses from participants in the present study, who noted that vegetable and whole-grain options have begun to pique the interest of their customers. However, more effort is needed to make these choices economically accessible and more competitive in terms of sensory appeal [37] and satiety among dining-out-food choices [38]. A domestic cross-sectional study suggested that the use of spices and herbs with vegetables encourages vegetable consumption among low-income populations [39]. Meanwhile, minimal incentives have also been found effective in encouraging the purchase of vegetable-rich meals among financially disadvantaged populations in interventions conducted in other countries, like Tokyo [40]. Finally, sugar reduction was the least supported concept by participants. In restaurants, sugar consumption primarily comes from sugar-sweetened beverages (SSBs), not food; however, reducing added sugar in beverages faces challenges due to a longstanding preference for sugary drinks in the United States, as indicated by a prospective cohort study from the Women’s Health Initiative, which tracked participants for a median of 20.9 years to analyze their intake of SSBs [41]. Research and policy efforts to reduce sugar consumption from restaurant sources have been implemented across jurisdictions outside of Baltimore, but these measures mostly apply to chain restaurants, rather than IORs [42,43].

Furthermore, owners’ insights into Baltimore’s local African American food culture have deepened our understanding of healthy food choices and practices, contributing to the development of an intervention acceptable by community members. For example, many menus at participating African American-owned restaurants feature a wide range of non-fried vegetable sides with African origins (e.g., collard greens, yams, black-eyed peas) that are seasoned with spices and herbs instead of salt [44]. This approach can be paired with a grilled entrée to create a healthy meal as part of the intervention. Similarly, the integration of local and community culinary traditions with healthy eating initiatives has been successfully implemented in other restaurant interventions, such as those conducted in Latinx restaurants in El Paso, Texas [45], and in Asian and Pacific Islander restaurants in San Diego, California [46].

Regarding the study’s limitations, although we adopted a framework with an established health standard, the NEMS-R tool is outdated regarding nutrition, having been published in 2007 with no revisions to date. This tool categorizes diet sodas as a “healthy” substitute for SSBs because they use no-calorie sweeteners. However, the healthfulness of low- and no-calorie sweeteners is still under debate [47]. We recommend adhering to the latest DGA, which suggest using diet soda solely for reducing added sugar intake and aiding in weight management [5]. We were unable to collect food recipes from all IORs or examine the actual nutritional content of the menu items. This limitation was due to the large number (nearly 1200) of food and beverage items across 14 IORs, compounded by the logistical and budgetary challenges of collecting food samples for laboratory testing. Additionally, we did not consider portion size when determining if a dish was healthy. However, most owners expressed that they use standard portioning during preparation, such as serving 6 ounces of ribeye steak entrée, which is notably twice the recommended serving size by the American Heart Association [48]. These limitations suggest that updates to the current standards for measuring the nutritional environment of restaurants and regulating restaurant food operations are needed to reflect the latest dietary guidelines.

Another limitation was the potential self-selection bias in our sampling. First, restaurant owners agreed to participate, and those that did might have been more health-conscious than others. Additionally, our sampling was not systematic; we initially relied on restaurants with whom we had pre-existing relationships. We observed that sit-down-restaurant owners, often located outside HFPA/low-income neighborhoods, were more responsive to conducting interviews compared to smaller carryout owners, potentially owing to the larger staffing capacity. Consequently, we failed to achieve an equal sample representation from HFPA/low-income neighborhoods and non-HFPA/low-income neighborhoods.

To mitigate the effects of this sampling bias, we performed a statistical analysis comparing two subgroups: restaurants located in HFPA/low-income neighborhoods and those situated outside of these areas. Our analysis showed no significant difference in the availability of healthy menu offerings between restaurants in these two groups, possibly due to the small sample size. A larger sample might have found significant differences. However, we achieved data saturation with our sample size in the qualitative analysis, which helped ensure the depth and breadth of the insights we gathered. Although the quantitative results show no significant differences, the qualitative depth bolsters the reliability of our findings within the studied context.

A strength of the study lies in its use of a mixed-methods design. The validity and reliability of identifying healthy restaurant offerings were enhanced through data triangulation from both observations and IDIs. Additionally, we employed a community-based approach in this formative research to ensure that the intervention strategies developed for the FRESH RCT were aligned with stakeholders’ interests and respectful of city culture, allowing for successful implementation in local IORs and beyond. Echoing the success of a Seattle-based intervention where a restaurant owner collaborated with health workers to create a cost-effective, diabetes-friendly menu [49], our study reinforces the notion that IORs are often keen to contribute positively to community health [18]. Future studies should further explore effective approaches to engage restaurant owners, especially those serving minority and low-income communities, in co-developing interventions to facilitate equitable access to healthy food in marginalized communities.

## 5. Conclusions

This formative research, utilizing a mixed-methods design and a community-based approach, assessed the availability of healthy restaurant offerings and identified feasible intervention points for the multi-site, multi-level RCT (“FRESH”). Our study indicates a need for interventions in IOR settings to increase the availability of healthy menu options as recommended by the DGA, which are also consistently recognized by owners, including non-fried lean proteins and non-starchy vegetables. Interventions should consider promoting healthy cooking methods agreed upon by owners, such as using low-cost vegetable oils instead of saturated fats and a blend of salt and seasonings to reduce sodium. Food policymakers should consider these factors when developing and expanding restaurant programs to combat the increasing trend of diet-related chronic diseases from FAFH sources. Additionally, further research efforts should explore feasible and sustainable changes from the customers’ perspective to facilitate them in making healthy choices when dining out.

## Figures and Tables

**Table 1 nutrients-16-01524-t001:** Descriptive characteristics of IORs in Baltimore, Maryland (*N* = 14).

Characteristics	*N* (%)
Restaurant type	data
Carryout restaurant	7 (50)
Sit-down restaurant	6 (43)
Food market stall	1 (7)
Sell alcohol	4 (29)
African American-owned	8 (57)
Located in HFPAs	5 (36)
Located in a neighborhood with a % of the population living below the poverty level	
≤5%	4 (28)
5–15%	4 (28)
≥15%	6 (44)
Located in a neighborhood with a % African American population	
≤10%	2 (14)
10–80%	6 (43)
≥80%	6 (43)

**Table 2 nutrients-16-01524-t002:** Differences in percentage of healthy menu offerings by HFPA/low-income neighborhood subgroups in 14 selected Baltimore IORs.

Characteristics	Total (*N* = 14)	HFPA/Low-Income (*N* = 6)	Non-HFPA/Low-Income (*N* = 8)	*p* ^1^
Healthy food ^2^	17.6 (14.3)	18.2 (17.0)	17.2 (13.2)	0.90
Poultry	5.1 (5.2)	7.3 (6.7)	3.5 (3.4)	0.23
Seafood	4.9 (5.5)	3.8 (3.5)	5.8 (6.8)	0.50
Dark-green vegetables	3.9 (3.5)	2.9 (2.6)	4.6 (4.1)	0.37
Starchy vegetables	1.4 (1.8)	1.6 (2.5)	1.2 (1.3)	0.73
Red and orange vegetables	0.8 (1.1)	1.0 (1.3)	0.6 (1.0)	0.58
Whole grains	0.6 (0.9)	0.8 (1.3)	0.4 (0.6)	0.43
Fruits	0.3 (0.7)	0.3 (0.8)	0.3 (0.6)	0.99
Soy products	0.3 (0.9)	0.0	0.6 (1.2)	0.19
Beans, peas, and lentils	0.2 (0.6)	0.3 (0.8)	0.2 (0.6)	0.74
Healthy beverages	26.0 (18.7)	26.1 (15.5)	26.0 (21.8)	0.99
Sugar-free beverages ^3^	14.5 (11.3)	11.5 (9.4)	16.8 (12.6)	0.39
Water	7.1 (7.4)	10.5 (8.5)	4.6 (5.7)	0.18
100% juice	4.4 (8.9)	4.0 (6.4)	4.6 (10.8)	0.90

All data are reported as mean (standard deviation). The percentages (%) presented here are calculated as the count of each categorical healthy food or beverage item divided by the count of total food or beverage items, multiplied by 100%. ^1^
*t*-test; *p* < 0.05 is considered statistically significant. ^2^ All food groups categorized as healthy are prepared in non-fried forms. ^3^ Sugar-free beverages include diet sodas such as Diet Coke, Sprite Zero, Diet Pepsi, Diet Root Beer, and Diet Dr. Pepper, as well as plain tea and coffee without added sugar.

**Table 3 nutrients-16-01524-t003:** Key themes, attitudes, reasons given, and suggestions from IOR owners for how they could make their foods healthier.

Themes	Attitude	Reasons Given	Suggestions
Using Healthier Cooking Fats	Positive	Customers will not notice the change in used oils.	Use a blend of olive oil and a less expensive oil to cut costs.
Salt Reduction and Favoring Seasoning	Mixed	Customers with hypertension want less salt; Customers would notice the change in taste.	Incorporating seasonings, herbs, and spices to replace a portion of the salt.
Adopting Non-Frying Cooking Methods	Mixed	Health-conscious customers want food prepared via non-frying methods; Most customers always choose fried options.	Using an air fryer instead of frying in oil.
Offering More Vegetables	Mixed	Some customers ask for vegetarian options; Meat dishes are more popular overall.	Introducing vegetable juices; Making vegetables the default side option.
Increasing Whole-Grain Options	Mixed	Fewer customers choose whole-wheat bread over white bread.	Offering whole-grain wraps; Introducing quinoa.
Sugar Reduction or Using Sweeteners	Negative	Customers crave sweet drinks; Customers would notice the change in taste; Sugary drinks like half-and-half (a mix of sweet tea and lemonade) are culturally significant in Baltimore.	Using brown sugar instead of white sugar in recipes. No refills for sweetened drinks.

## Data Availability

All data generated or analyzed during this study are included in this published article. The raw data supporting the conclusions of this article will be made available by the authors on request due to privacy reasons.

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
