# Peer review of "The Quality of Menu Offerings in Independently Owned Restaurants in Baltimore, Maryland: Results from Mixed-Methods Formative Research for the FRESH Trial"

_nutrients, 2024, doi:10.3390/nu16101524_

Round 1

Reviewer 1 Report

Comments and Suggestions for Authors

1.       I suggest replacing "Diet" in the title with "Menu Offerings", because the presented results refer to the menu, not to the diet.

2.       In lines 62-63, I propose to remove "like supermarkets", because supermarkets are not a good source of healthy food.

3.       In line 108, the ArcGIS abbreviation requires additional explanation.

4.       Why were desserts omitted from the study? This requires additional explanation.

5.       I propose to remove subsection 2.6, because it shows that outsiders evaluate menus in Baltimore restaurants, what does not reflect well on the exclusion of the local population as authors of the research.

6.       In the title of table 2, I propose adding "selected" before IORs.

7.       Chapter 3.2 requires more detailed description.

8.       What does the P column in table 2 mean? Is that p-value of Chi-square? If so, recalculation is required because the values included in the table are not correct.

9.       In tables 1-3, percentages do not always add up to 100.

10.   There is no statistical analysis for the results in table 3.

11.   Chapter 3.3 presents an unscientific description of qualitative research. A comparison to results obtained by other authors in other cities and countries would be useful here.

12.   There is a misunderstanding in lines 388 and 389, after all, white bread is made from both wheat and rye. This sentence requires rewording.

13.   There is an error in lines 463-465 because the research results do not suggest that the owners' opinion contrasts with the WHO. The sentence needs to be rephrased.

14.   The study was carried out in only 14 restaurants, so it was possible to examine the recipe composition of all dishes very carefully, without the need for laboratory analyses. This omission by the authors also should be taken into account in the conclusions.

15.   PS. The solution seems to be the introduction of a sugar tax, a significant increase in the price of disposal of fat used for frying, and an increase in the price of red meat as it was done I Europe.

Author Response

1.  I suggest replacing "Diet" in the title with "Menu Offerings", because the presented results refer to the menu, not to the diet.  

We agree that “Menu Offerings” better reflects our research focus. We have replaced “Diet” in the title with “Menu Offerings”. 

2.  In lines 62-63, I propose to remove "like supermarkets", because supermarkets are not a good source of healthy food.  

New lines 62-65: We appreciate your proposal. In the context of the US, supermarkets are considered healthier food sources because they often offer a variety of fresh produce not available in smaller retail settings. To avoid misinterpretation, we have modified the definition by referring to the official definition of a Healthy Food Priority Area from the local government. 

3. In line 108, the ArcGIS abbreviation requires additional explanation.  

New line 112: We clarified the use of ArcGIS by specifying it as a geographic information system software developed by Esri, used for mapping and analyzing spatial information. We have now included the version and release date: ArcGIS (June 2022, version 1.9).        

4. Why were desserts omitted from the study? This requires additional explanation.  

New lines 160-161: We have clarified that desserts were excluded because these items are not categories listed in the NEMS-R protocol for menu item counting.         

5. I propose to remove subsection 2.6, because it shows that outsiders evaluate menus in Baltimore restaurants, what does not reflect well on the exclusion of the local population as authors of the research.  

We appreciate your careful consideration. However, research reflexivity is a crucial component in qualitative research. We believe it is important to retain Section 2.6 to reflect on how our background (termed “positionality”) influences our perspectives and interactions during the research process. 

To address concerns about being perceived as “outsiders,” we have included the range of our team members’ familiarity with the study context; they have lived and worked at the study site for anywhere from 2 to 30 years.  

Darwin Holmes, A. G. (2020). Researcher Positionality—A Consideration of Its Influence and Place in Qualitative Research—A New Researcher Guide. Shanlax International Journal of Education, 8(4), 1–10. https://doi.org/10.34293/education.v8i4.3232 

6. â€¯In the title of table 2, I propose adding "selected" before IORs.  

We have revised table 2 title based on your suggestion. 

7. Chapter 3.2 requires more detailed description.  

New lines 215-221: We have expanded the description in section 3.2. 

The section is now divided into two paragraphs to address the two semi-research questions previously stated in the introduction: What is the current availability of healthy foods and beverages in IORs in Baltimore? And how does this availability vary by neighborhood health food access and income level? 

The first question is addressed descriptively, detailing the number and percentage of healthy items observed and the number and percentage of IORs offering these items. The second question is addressed using an interpretable table that displays the p-value of the t-test. 

8. What does the P column in table 2 mean? Is that p-value of Chi-square? If so, recalculation is required because the values included in the table are not correct.  

Thank you for your query. The p-values reported in Table 2 are from t-tests, based on recalculations. New lines 196-199: We have also updated Section 2.8 on data analysis accordingly. These recalculations accurately compare the mean percentages of healthy menu offerings between two independent restaurant samples within different neighborhood types.  

9. In tables 1-3, percentages do not always add up to 100.  

We reviewed and corrected all tables to ensure that percentages accurately reflect the data. In Table 1, categorical variables now sum to 100%. In Table 2, we have added a footnote stating: “The percentages (%) presented here are calculated as the count of each categorical healthy food or beverage item divided by the total count of food or beverage items, multiplied by 100%.” This clarification explains why sub-categorical totals do not sum to 100% but rather add up to the percentage of the main categories (healthy food and healthy beverages). 

10. There is no statistical analysis for the results in table 3.  

Table 3 has been removed to streamline the presentation of results. 

11. Chapter 3.3 presents an unscientific description of qualitative research. A comparison to results obtained by other authors in other cities and countries would be useful here. 

Our description of qualitative research in Section 3.3 presents a thematic finding regarding what food owners perceive as healthy on their menus. This approach aligns with established methods in qualitative analysis as outlined by Burnard, P., Gill, P., Stewart, K., Treasure, E., & Chadwick, B. (2008) in their article “Analysing and presenting qualitative data,” published in the British Dental Journal.  

Burnard, P., Gill, P., Stewart, K., Treasure, E., & Chadwick, B. (2008). Analysing and presenting qualitative data. British Dental Journal, 204(8), 429–432. https://doi.org/10.1038/sj.bdj.2008.292 

New lines 487-492: Comparisons of our results with those from other studies in different cities and countries have been addressed in the discussion section.  

12. There is a misunderstanding in lines 388 and 389, after all, white bread is made from both wheat and rye. This sentence requires rewording.  

New lines 312-314: The sentence regarding the bread flour type has been rephrased for clarity and accuracy.  

13. There is an error in lines 463-465 because the research results do not suggest that the owners' opinion contrasts with the WHO. The sentence needs to be rephrased.  

New lines 483-487: We rephrased the sentences and clarified how owners’ opinions regarding the healthfulness of processed meat do not align with WHO recommendations. 

14. The study was carried out in only 14 restaurants, so it was possible to examine the recipe composition of all dishes very carefully, without the need for laboratory analyses. This omission by the authors also should be taken into account in the conclusions.  

New lines 539-542: This analysis of recipes is beyond the scope of our study. We have documented a total of 1,019 food items and 174 beverage items across 14 IOR samples, as presented in Section 3.2. Ingredients such as salt and oil are not always added according to a fixed recipe; therefore, accurately assessing their quantities would require laboratory analysis. Conducting such comprehensive lab analyses for the full menu of 14 restaurants is prohibitively expensive. 

15. PS. The solution seems to be the introduction of a sugar tax, a significant increase in the price of disposal of fat used for frying, and an increase in the price of red meat as it was done I Europe. 

Thank you for your suggestion regarding food policy. As these policies take effect and are considered for implementation as interventions, we plan to incentivize nudge owners to purchase healthier ingredients (e.g., canola oils, vegetables, lean proteins). Additionally, we will offer subsidies and lower prices for non-fried, vegetable-rich meals to encourage customer purchases. We have included a reference in the discussion section about an intervention in Tokyo that uses incentives to encourage customers to buy vegetable-rich meals (ref 40). 

Reviewer 2 Report

Comments and Suggestions for Authors

This mixed method study employed qualitative interviews and descriptive statistics to evaluate perceptions owners of independently owned restaurants about healthy food and customer acceptance of healthier menu options. The methods and results address the three research questions.

Title

The use of the word “improving” implies the results are focused on the actual intervention rather than just the interviews with owners. Suggest deleting the word “improving” from the title.

Instead of multi-methods, suggest using mixed methods to describe study type.

Introduction

Line 1: The data analyzed in the reference 1 study is from the late 1970s and 1990s. Is there a more current reference that you could site?  If not, I don’t think it is accurate to say, “In recent years” because the data is from decades ago. Suggest revising that sentence/phrase.

Materials and Methods

Line 92: Instead of multi-methods, suggest using mixed methods to describe study type.

Line 92: Explain formative research—I am assuming the findings will information the intervention of the FRESH trial. If so, include sentence about that. If not, include sentence regarding purpose of this formative research.

Line 96: Ref 23. NEMS-R protocol—is there a year, version to add for this tool?

Line 108: Recruitment strategy: ArcGIS—is there a year, version to add for this product?

Line 113: Is first person permitted by this journal? If not, revised this sentence (we…) and check rest of manuscript for the same needed change.

Line 114: Add sentence about how data saturation was defined/determined. Was a power calculation done?

Hennink M, Kaiser BN. Sample sizes for saturation in qualitative research: A systematic review of empirical tests. Soc Sci Med. 2022 Jan;292:114523.

Line 134: Does “their” refer to Delivery App websites or IOR websites. The way it is written it implies Delivery App website. If it is IOR website, suggest changing “their” to “IOR.”

Line 137: 2007 conference, that’s 17 years ago. Is that not more current guideline?

Lines 175-176: ATLAS.ti software (DATE, version 23.4)—add release date for that version of the software.

Lines 185-186: Using R software (DATE, version 4.2.1)—add release date for that version of the software.

Lines 185-188: For Chi Square test, add significance level, was it p = .05?

Results

Line 194: Given alcohol was not a scored food item, why are you not including it in the opening paragraph? I think the more significant item from the table to highlight in the text is that 50% of the IOR were carry out restaurants.

Line 202: change observed from menus to complied from menus.

Lines 203: Instead of Table 2 in parenthesis, suggest stating that Table 2 provides a comparison on availability of healthy food items at HFPA vs. non-HFPA—or similar statement.

Line 206: Ditto re Table 3. Add sentence stating what Table 3 illustrates.

Lines 210-211: The most frequently available healthy 210 beverages were sugar-free beverages (17.8%), provided by 10 IOR—do more current tools classify diet sodas as healthy?

Line 225: Change include to included.

Line 252: Select synonym for significance. Only use when data has been found to be statistically significant.

Line 274: You might add to the methods that filler words (um, like, you know, etc.) were removed from transcripts. Then, you could remove the “likes” from this quote.

Line 291: See note about line 274 re “you know.”

Line 297: Were the components of change compiled from the interview data? If so, state that information.  If not, explain how the components of change were identified.

Table 4: Change Vegetable to Vegetables.

Line 446: Recommend changing “established health standards” to “established standards for healthy foods.”  That way it does not imply that the restaurants were not compliant with public health standards such as sanitary practices.

Line 484: Least supported [concept]… add the word concept of another word.

Line 487: Is there a more recent study than the one conducted in reference 35. The strength of the argument would be stronger with a more recent data set.

Line 495: Check the accuracy of the statement that “non-fried vegetable sides with African origins (e.g., collard greens, yams, cabbages, green beans)” and add a reference. Do these foods have African origins or their roots in American slavery/southern US?

Lines 521-522: The Methods states that data saturation was achieved, which helps to put the sample size into perspective.

Hennink M, Kaiser BN. Sample sizes for saturation in qualitative research: A systematic review of empirical tests. Soc Sci Med. 2022 Jan;292:114523.

Line 539- 542: “Statistics revealed that healthy options most commonly available are non-fried poultry, seafood, non-starchy vegetables, and sugar-free beverages, with no significant differences in the availability of these items between restaurants in HFPA/low-income neighborhoods and those in other areas.”  I think you are over selling this by saying “statistics” reveal—the statistical tests used (descriptive and Chi Square) are not rigorous analysis.  Also, recommended taking sugar-free beverages out as an example.  The inclusion of them as a healthy options reflects the age of the tool use. Today, they are not a healthy food but an item to enjoy in moderation.

References

Inconsistent formatting of references, ensure all comply with journal style requirements.

Ref 24. Add the rest of the citation.

Some references are older, more recent references would strengthen the article:

·       References 1, 30, 35, 41

Comments on the Quality of English Language

Included with other comments.

Author Response

Title 

The use of the word “improving” implies the results are focused on the actual intervention rather than just the interviews with owners. Suggest deleting the word “improving” from the title. 

Instead of multi-methods, suggest using mixed methods to describe study type. 

Response: We updated the title to “Menu Offerings Quality in Independently Owned Restaurants in Baltimore, Maryland: Results from Mixed-Methods Formative Research for the FRESH Trial” per both peer reviewers’ suggestions. 

Introduction 

Line 1: The data analyzed in the reference 1 study is from the late 1970s and 1990s. Is there a more current reference that you could site?  If not, I don’t think it is accurate to say, “In recent years” because the data is from decades ago. Suggest revising that sentence/phrase. 

Response: New line 39: We revised it to a more accurate statement “over the past three decades” and cited a more recent reference from a 2018 USDA report. 

Materials and Methods 

Line 92: Instead of multi-methods, suggest using mixed methods to describe study type. 

Response: New lines 80, 96, 567, 580: We adopted the term “mixed-methods” throughout the document to better describe our study type.  

Line 92: Explain formative research—I am assuming the findings will information the intervention of the FRESH trial. If so, include sentence about that. If not, include sentence regarding purpose of this formative research. 

Response: New lines 84-86: We added a sentence to clarify the purpose of formative research in informing the FRESH trial intervention. 

Line 96: Ref 23. NEMS-R protocol—is there a year, version to add for this tool? 

Response: New line 100: We added the year of the NEMS-R protocol, April 2007. The protocol can be accessed on the official NEMS website (https://nems-upenn.org/tools/). To our knowledge, it has not been officially revised or updated into different versions. Our review of other publications using NEMS-R did not reveal any versions of this tool. 

Line 108: Recruitment strategy: ArcGIS—is there a year, version to add for this product? 

Response: New line 112: The year and version for the ArcGIS Online software used have been added as June 2022, Version 1.9.   

Line 113: Is first person permitted by this journal? If not, revised this sentence (we…) and check rest of manuscript for the same needed change. 

Response: We double-checked the guidelines and can confirm that Nutrients does not restrict the use of first-person narrative. 

Line 114: Add sentence about how data saturation was defined/determined. Was a power calculation done? 

Hennink M, Kaiser BN. Sample sizes for saturation in qualitative research: A systematic review of empirical tests. Soc Sci Med. 2022 Jan;292:114523. 

Response: New line 119-121: Thank you for the suggested reference. We clarified that data saturation was determined by the data collectors when no new themes or information emerged from subsequent interviews. No power calculation was conducted, as it is not feasible to foresee the themes that will emerge and to determine the necessary sample size in advance. This approach follows the qualitative research guidelines outlined by Sandelowski, M. (1995) in “Sample size in qualitative research,” published in Research in Nursing & Health, 18(2), 179-183. https://doi.org/10.1002/nur.4770180211 

Line 134: Does “their” refer to Delivery App websites or IOR websites. The way it is written it implies Delivery App website. If it is IOR website, suggest changing “their” to “IOR.” 

Response: New line 141: We clarified the reference to “their” specifically pointing to “IOR” websites. 

Line 137: 2007 conference, that’s 17 years ago. Is that not more current guideline? 

Response: We acknowledge concerns about the currency of the guidelines cited in “Performance Standards for Restaurants: A New Approach to Addressing the Obesity Epidemic” by Deborah Cohen, Rajiv Bhatia, Mary T. Story, et al., published in 2013. These guidelines stem from a 2012 conference supported by the NIH/NIMHD in Santa Monica, California, where thirty-eight US experts reviewed nutrition standards, including the DGA and IOM guidelines, and initiatives like ‘Por Vida’ and NRA’s Kids LiveWell (p2). They developed standards to help restaurants promote obesity prevention and healthier dietary choices aligned with the DGA. We have not found more recent guidelines that match our menu analysis goals as closely as these do. 

Lines 175-176: ATLAS.ti software (DATE, version 23.4)—add release date for that version of the software. 

Response: New line 185: Added December 2023. 

Lines 185-186: Using R software (DATE, version 4.2.1)—add release date for that version of the software. 

Response: New line 196: Added June 2022. 

Lines 185-188: For Chi Square test, add significance level, was it p = .05? 

Response: New lines 196-198: Yes. Added significance level to both Methods and the footnote of Table 2. 

Results 

Line 194: Given alcohol was not a scored food item, why are you not including it in the opening paragraph? I think the more significant item from the table to highlight in the text is that 50% of the IOR were carry out restaurants. 

Response: New line 205: Based on your advice, we removed the mention of alcohol service and highlighted the prevalence of carryout restaurants within our IOR sample. 

Line 202: change observed from menus to complied from menus. 

Response: New line 212: We changed “observed” to “compiled”. 

Lines 203: Instead of Table 2 in parenthesis, suggest stating that Table 2 provides a comparison on availability of healthy food items at HFPA vs. non-HFPA—or similar statement. 

Response: New line 228: We added a sentence at the beginning of paragraph 2 in Chapter 3.2 that clearly states: Table 2 presents a comparison of the healthy menu offerings between two samples of IORs in HFPA/low-income neighborhoods and non-HFPA/low-income neighborhoods.  

Line 206: Ditto re Table 3. Add sentence stating what Table 3 illustrates. 

Response: Following another reviewer’s recommendation, we have removed Table 3. The descriptive data previously found in Tables 2 and 3 are now consolidated and summarized in the opening paragraph, with Table 2 exclusively presenting the t-test analysis. 

Lines 210-211: The most frequently available healthy 210 beverages were sugar-free beverages (17.8%), provided by 10 IOR—do more current tools classify diet sodas as healthy? 

Response: We categorized diet sodas as healthy drink alternatives based on their low-calorie content and absence of added sugars, following the NEMS-R protocol. New lines 533-539: However, we acknowledged the tool’s outdated status and addressed this limitation in the discussion section. According to the current Dietary Guidelines for Americans 2020-2025, low- and no-calorie sweeteners are recommended as substitutes for added sugars to help reduce calorie intake and aid in weight management. The long-term health impacts of low and no-calorie sweeteners are still under debate. Nevertheless, the FDA considers these sweeteners safe, and they can be effective strategies for weight management. This viewpoint is supported by recent research, including a systematic review and meta-analysis by McGlynn ND, Khan TA, Wang L, et al., published in JAMA Network Open (2022), which investigates the association of low- and no-calorie sweetened beverages as replacements for sugar-sweetened beverages with body weight and cardiometabolic risk. 

McGlynn ND, Khan TA, Wang L, et al. Association of Low- and No-Calorie Sweetened Beverages as a Replacement for Sugar-Sweetened Beverages With Body Weight and Cardiometabolic Risk: A Systematic Review and Meta-analysis. JAMA Netw Open. 2022;5(3):e222092. Published 2022 Mar 1. doi:10.1001/jamanetworkopen.2022.2092 

Line 225: Change include to included. 

Response: New line 249: Grammar revised. 

Line 252: Select synonym for significance. Only use when data has been found to be statistically significant. 

Response: New line 276: Changed word “significance” to “crucial roles”. 

Line 274: You might add to the methods that filler words (um, like, you know, etc.) were removed from transcripts. Then, you could remove the “likes” from this quote. 

Response: New lines 135-136: Thank you for your suggestions. We have removed filler words from the transcripts and included a sentence in Section 2.4 of the Methods to specify this cleanup process. 

Line 291: See note about line 274 re “you know.” 

Response: We have cleaned filler words. 

Line 297: Were the components of change compiled from the interview data? If so, state that information.  If not, explain how the components of change were identified. 

Response: New line 322: We have clarified that the themes are organized by each component of change as outlined in the questions from the IDI guide. This structure was directly informed by the interview data. 

Table 4: Change Vegetable to Vegetables. 

Response: New Table 3: Grammar revised. 

Line 446: Recommend changing “established health standards” to “established standards for healthy foods.”  That way it does not imply that the restaurants were not compliant with public health standards such as sanitary practices. 

Response: New line 468: Thank you for your thoughtful suggestion. We revised the terminology to “established standards for healthy meals”. 

Line 484: Least supported [concept]… add the word concept of another word. 

Response: New line 513: Added the word “concept”. 

Line 487: Is there a more recent study than the one conducted in reference 35. The strength of the argument would be stronger with a more recent data set. 

Response: New line 516-518: We have updated our referred data set (ref 41): a prospective cohort from the Women’s Health Initiative, followed for a median of 20.9 years until 2020.  

Zhao, L., Zhang, X., Coday, M., Garcia, D. O., Li, X., Mossavar-Rahmani, Y., Naughton, M. J., Lopez-Pentecost, M., Saquib, N., Shadyab, A. H., Simon, M. S., Snetselaar, L. G., Tabung, F. K., Tobias, D. K., VoPham, T., McGlynn, K. A., Sesso, H. D., Giovannucci, E., Manson, J. E., Hu, F. B., … Zhang, X. (2023). Sugar-Sweetened and Artificially Sweetened Beverages and Risk of Liver Cancer and Chronic Liver Disease Mortality. JAMA, 330(6), 537–546. https://doi.org/10.1001/jama.2023.12618 

Line 495: Check the accuracy of the statement that “non-fried vegetable sides with African origins (e.g., collard greens, yams, cabbages, green beans)” and add a reference. Do these foods have African origins or their roots in American slavery/southern US? 

Response: New line 526: We have verified the origins of the foods mentioned with “William Frank Mitchell. African American Food Culture. Westport, Connecticut: Greenwood Press; 2009.” Based on this source, while collard greens and yams do have African origins, cabbages, and green beans, although prevalent in black-owned restaurants in Baltimore, were not introduced to the American continent through the transatlantic slave trade. We have revised the statement accordingly to reflect this accuracy. 

Lines 521-522: The Methods states that data saturation was achieved, which helps to put the sample size into perspective. 

Hennink M, Kaiser BN. Sample sizes for saturation in qualitative research: A systematic review of empirical tests. Soc Sci Med. 2022 Jan;292:114523. 

Response: New lines 563-566: Thank you for highlighting this aspect. We have included a discussion on data saturation to address and justify the limitations related to the study’s small sample size. 

Line 539- 542: “Statistics revealed that healthy options most commonly available are non-fried poultry, seafood, non-starchy vegetables, and sugar-free beverages, with no significant differences in the availability of these items between restaurants in HFPA/low-income neighborhoods and those in other areas.”  I think you are over selling this by saying “statistics” reveal—the statistical tests used (descriptive and Chi Square) are not rigorous analysis.  Also, recommended taking sugar-free beverages out as an example.  The inclusion of them as a healthy options reflects the age of the tool use. Today, they are not a healthy food but an item to enjoy in moderation. 

Response: New lines 582-585: Thank you for your critical feedback. We acknowledge that our phrasing regarding the quantitative results may have overstated the findings given the small sample; hence, the focus of our conclusion has been adjusted to emphasize future implications for interventions in IOR settings, rather than overstating our initial findings. We also removed sugar-free beverages here as a healthy beverage example due to the uncertainty around the healthfulness of sweeteners. 

References 

Inconsistent formatting of references, ensure all comply with journal style requirements. 

Ref 24. Add the rest of the citation. 

Some references are older, more recent references would strengthen the article: 

  •        References 1, 30, 35, 41

Response: We have addressed the issue of inconsistent formatting and ensured that all references now comply with the journal’s style requirements. We have completed the citation for Reference 24, which is now labeled as Reference 26. Reference 1 has been updated to a more recent report. Reference 35, now Reference 41, has been updated to a 2023 publication. Reference 30, now Reference 35, has been updated to a 2023 report. We were unable to find a more recent source for a restaurant-owner-initiated program that improves menu healthfulness, similar to what was cited in Reference 41, now Reference 49.